# The impact of brominated flame retardants exposure on serum total bilirubin: A cross-sectional analysis

**Shanshan Huang[1], Tong Lin[1], Jialu Chen[1], Fen Zhou[1], Junjie Yang[1], Haiyan Mao[1]\*,
Zhouxin Yang[2]\***

**1** Department of Critical Care Medicine, Ningbo Medical Center Lihuili Hospital, Ningbo, China, **2** Zhejiang Key Laboratory of Geriatrics and Geriatrics Institute of Zhejiang Province, Zhejiang Hospital, Hangzhou, China

\* maomao2003678@163.com (HM); yangzhouxin@hotmail.com (ZY)

## Abstract

### Background

Brominated flame retardants (BFRs) are harmful, bioaccumulative, and persistent environmental pollutants, posing significant health risks. Elevated bilirubin levels can cause neurotoxicity and damage to the heart, liver, kidneys, and other organs. This study utilizes National Health and Nutrition Examination Survey (NHANES) data to investigate the association between exposure to BFRs and total bilirubin (TB) levels in adult participants.

### Methods

Based on data from the NHANES 2007-2016, TB levels were divided into tertiles. Spearman's rank correlation was used to analyze the relationship between individual BFRs and TB levels. Weighted linear regression analysis, restricted cubic splines (RCS), and stratified analysis were conducted to assess the correlation between individual BFRs and TB levels. Weighted quantile sum (WQS) regression and quantile-based g-computation (QGC) analysis were used to comprehensively evaluate the impact of BFRs exposure on serum TB levels.

### Results

The study included 5831 participants. The results showed that PBB153, PBDE17, PBDE47, PBDE85, PBDE99, PBDE100, PBDE209, and PBDE183 were significantly correlated with TB levels ($p < 0.05$), with PBDE183 having the highest Spearman rank correlation coefficient of 0.292. After adjusting for confounding factors, most BFR remained significantly positively correlated with TB, while PBDE153 (β: - 0.031, 95%CI: - 0.317, 0.255, $p = 0.829$) and PBDE66 (β: 0.285, 95%CI: - 0.208, 0.777, $p = 0.253$) were not statistically significant. RCS analysis indicated that PBDE153 concentration had a significant U-shaped correlation with TB ($p < 0.05$), while PBDE17, PBDE99, PBDE154, and PBDE209 had an inverted "J"-shaped correlation ($p < 0.05$). PBB153, PBDE66, PBDE85, and PBDE183 also exhibited significant nonlinear S-shaped correlations with TB ($p <$

**Data availability statement:** This study analyzed publicly available datasets, which can be found at 10.6084/m9.figshare.28466081.

**Funding:** Zhejiang Provincial Natural Science Foundation of China (LY21H150002) and Scientific Research Fund of Zhejiang Province Education Department (Y202043331)

**Competing interests:** The authors have declared that no competing interests exist.

0.05). After stratification by age and gender, most individual BFR remained significantly positively correlated with TB levels ($p < 0.05$). WQS regression and QGC analysis indicated that mixed BFRs exposure was positively correlated with TB levels (β: 0.553, 95%CI: 0.384, 0.722, $p < 0.001$ and β: 1.060, 95%CI: 0.587, 1.532, $p < 0.001$), with PBDE183 contributing the most.

## Conclusions

BFRs exposure is significantly positively correlated with TB levels, further suggesting the potential health impact of BFRs exposure on humans.

## 1. Introduction

Protecting human health from the effects of environmental exposure has always been a serious public health concern. According to the World Health Organization, in 2016, ecological pollution was responsible for an estimated 4.2 million deaths globally [1]. Although there is currently no statistical data on deaths caused by environmental pollution from brominated flame retardants (BFRs), evidence suggests that exposure to BFRs is closely associated with adverse health outcomes and may increase the risk of tumor-related mortality [2]. BFRs are harmful, bioaccumulative, and persistent pollutants found in various environmental matrices, posing a significant threat to human health [3].

BFRs include polybrominated diphenyl ethers (PBDEs) and polybrominated biphenyls (PBBs), among which PBDEs comprise 209 congeners [4]. They are widely used in commercial and industrial applications, including furniture, textiles, plastics, electronic devices, and building materials, to reduce flammability and decrease the risk of fires, meeting fire safety standards. Humans are continuously exposed to these flame retardants through various routes, including diet, breastfeeding, drinking water, and using products containing these substances [4–6]. Despite the banning of several flame retardants, these substances persist in the environment due to their slow degradation and tendency to bioaccumulate [7–9]. It has been reported that although the United States ceased the production of Polychlorinated Biphenyls (PCBs) in 1979, these substances remain persistent in the environment and can accumulate through breastfeeding and the food chain [10–12]. The toxicological effects of various BFRs are similar, with the liver, kidneys, and thyroid being the main target organs. They also have certain impacts on the endocrine, nervous, and reproductive systems [13,14].

The liver is a crucial organ involved in various physiological processes, including metabolism, volume regulation, immune function, endocrine signaling regulation, lipid homeostasis, and detoxification. Liver failure can lead to severe consequences [15,16]. Bilirubin, released from the breakdown of aging red blood cells, is a toxic byproduct of heme catabolism. It serves as a marker of liver dysfunction and reflects liver disease [17–19]. Additionally, bilirubin can cause neurological damage in newborns and promote inflammation that interferes with brain development, thus playing a role in the induction of neuropsychiatric disorders [20–22].

BFRs are closely associated with human health through daily exposure. However, the association between mixed or individual BFRs and total bilirubin (TB) levels has not been established. Therefore, this study utilizes data from the National Health and Nutrition Examination Survey (NHANES) to investigate the relationship between BFR exposure and TB levels in adult participants.

## 2. Materials and methods

### 2.1 Data source and study population

This study utilized data from the NHANES, which provides extensive information on the nutrition and health of the general U.S. population. The NHANES study was approved by the National Center for Health Statistics (NCHS) Research Ethics Review Board, and all survey participants signed informed consent forms. Detailed information about the NCHS Research Ethics Review Board approval can be accessed from the NHANES website (https://www.cdc.gov/nchs/nhanes/irba98.htm). The data were extracted from the NHANES database for the years 2007-2016, encompassing five consecutive survey cycles. The initial dataset included 50,588 participants. The following participants were excluded: 1) missing BFRs data (41,752); 2) missing TB data (20); 3) age < 20 years old (1,552); 4) missing covariates data (1,433). Fig 1 illustrates the specific screening process used in this study.

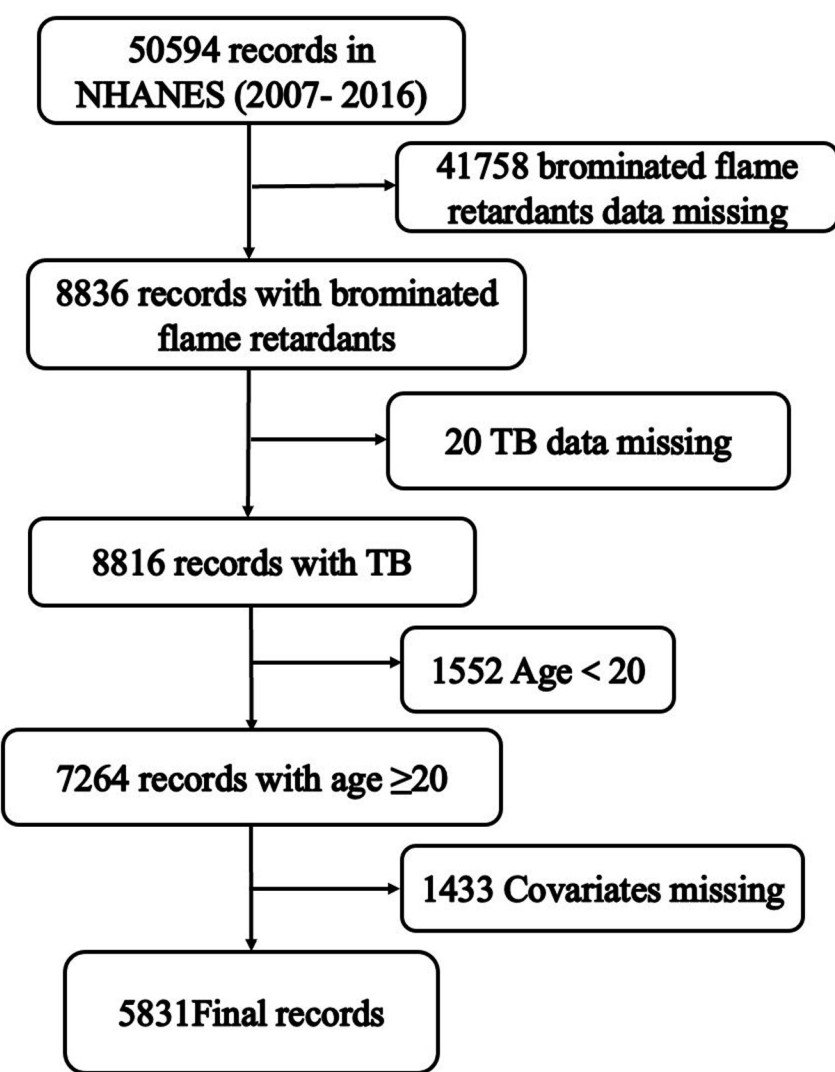

**Fig 1. Flow chart for the study.** (TB, total bilirubin).

## 2.2 Measurement of BFRs and TB

The NHANES database includes measurements of 12 types of BFRs: PBB153, PBDE100, PBDE17, PBDE28, PBDE66, PBDE47, PBDE85, PBDE99, PBDE153, PBDE183, PBDE209, and PBDE154. BFRs were analyzed using automated liquid-liquid extraction and purification, followed by target analyte analysis through isotope dilution gas chromatography with high-resolution mass spectrometry GC/IDHRMS [23]. TB was measured and quality-controlled using the timed-endpoint diazo method (Jendrassik-Grof) at the NHANES mobile examination centers [24]. Detailed sample collection and measurement methods for BFRs and TB can be found in the NHANES Laboratory Procedures Manual.

## 2.3 Covariates

The study included the following covariates: age (years), sex (male or female), race (Mexican American, non-Hispanic Black, non-Hispanic White, other Hispanic, other/multiracial), poverty-to-income ratio (PIR), education level (less than high school, high school, more than high school), body mass index (BMI, underweight, normal weight, overweight, or obese), white blood cell count (WBC), neutrophil count (Neu), platelet count (PLT), lymphocyte count (Lym), systemic immune-inflammation index (SII, calculated as PLT × Neu/ Lym), creatinine, alanine aminotransferase (ALT), aspartate aminotransferase (AST), alkaline Phosphatase (ALP), blood sodium level (Na), blood potassium level (K), smoking status (yes or no), drinking status (yes or no), hypertension (yes or no), and diabetes (yes or no).

## 2.4 Statistical methods

To reduce data variability, weighted methods were used in the analysis. Descriptive statistics were performed with continuous variables, including TB, grouped into tertiles. Continuous variables were analyzed using the Wilcoxon rank-sum test and reported as means and standard deviations. Categorical variables were analyzed using the chi-squared test with Rao & Scott's correction and presented as percentages. The Kolmogorov-Smirnov test was used for the initial normality assessment, followed by Spearman's rank correlation analysis to explore variable relationships.

Three models were established: Model 1 (unadjusted), Model 2 (adjusted for age, gender, race, PIR, education, and BMI), and Model 3 (adjusted for age, gender, race, PIR, education, BMI, smoking status, drinking status, hypertension, and diabetes). After log transformation of the BFRs, weighted generalized linear regression analysis, restricted cubic spline (RCS, with 4 knots), and subgroup analysis were used to study the association between in and TB levels.

Finally, based on Model 3, weighted quantile sum (WQS) regression was used for quantile regression to evaluate the overall effect of the BFRs mixture on TB. In this model, the data were randomly divided into a training set (30%) and a validation set (70%), and 1,000 bootstrap samples from the training dataset were used to calculate the regression coefficients. Quantile-based g-computation (QGC) analysis combined quantile regression with G-computation to assess the joint effect of BFRs exposure on TB, with 1,000 bootstrap iterations used to evaluate mixed slopes and overall model confidence. All data analyses were conducted using R version 4.4.1, with statistical significance set at $p < 0.05$.

## 3. Results

### 3.1 Baseline characteristics

This study ultimately included 5,831 participants. After weighting, 51.29% of the participants were female and 48.71% were male, with an average age of 46.97 years (Table 1). According

**Table 1. Weighted characteristics of the study participants by tertiles of TB.**

| Characteristics | Overall, N = 5831 | TB Tertiles Q1, N = 1982[1] | Q2, N = 1921[1] | Q3, N = 1928[1] | p-value[2] |
|---|---|---|---|---|---|
| BFRs (pg/g) | | | | | |
| PBB153 | 1.31 (0.93) | 1.37 (0.94) | 1.29 (0.90) | 1.26 (0.95) | <0.001 |
| PBDE17 | 28.01 (54.84) | 27.20 (45.78) | 29.23 (63.73) | 27.60 (53.34) | 0.034 |
| PBDE28 | 33.47 (29.88) | 34.60 (30.70) | 33.00 (29.18) | 32.85 (29.75) | 0.088 |
| PBDE47 | 1.71 (0.89) | 1.65 (0.85) | 1.74 (0.94) | 1.73 (0.86) | <0.001 |
| PBDE66 | 72.04 (61.08) | 72.18 (63.26) | 71.74 (60.91) | 72.20 (59.11) | 0.180 |
| PBDE85 | 3.20 (3.32) | 3.25 (3.15) | 3.13 (3.19) | 3.22 (3.59) | 0.028 |
| PBDE99 | 1.97 (3.73) | 2.05 (3.44) | 1.93 (3.90) | 1.92 (3.82) | <0.001 |
| PBDE100 | 19.66 (36.60) | 19.45 (31.11) | 19.81 (38.23) | 19.69 (39.73) | 0.025 |
| PBDE153 | 8.67 (6.48) | 8.82 (6.56) | 8.61 (6.42) | 8.58 (6.48) | 0.364 |
| PBDE154 | 159.93 (141.69) | 163.19 (140.35) | 158.62 (140.20) | 158.09 (144.43) | 0.131 |
| PBDE183 | 3.59 (3.82) | 3.66 (3.75) | 3.53 (3.73) | 3.56 (3.97) | 0.018 |
| PBDE209 | 35.08 (42.61) | 35.55 (40.15) | 34.86 (43.18) | 34.83 (44.35) | 0.007 |
| Age | 46.97 (16.82) | 46.38 (16.22) | 47.21 (16.98) | 47.30 (17.22) | 0.404 |
| Gender | | | | | <0.001 |
| Male | 48.71% | 33.90% | 45.23% | 66.42% | |
| Female | 51.29% | 66.10% | 54.77% | 33.58% | |
| Races | | | | | <0.001 |
| Mexican American | 8.22% | 9.19% | 8.16% | 7.35% | |
| Non-Hispanic Black | 11.01% | 13.71% | 10.98% | 8.44% | |
| Non-Hispanic White | 68.31% | 62.29% | 69.53% | 72.92% | |
| Other Hispanic | 5.54% | 6.60% | 5.07% | 4.99% | |
| Other/multiracial | 6.91% | 8.21% | 6.26% | 6.30% | |
| PIR | 3.00 (1.64) | 2.76 (1.64) | 3.05 (1.63) | 3.18 (1.64) | <0.001 |
| Education | | | | | 0.006 |
| High | 61.19% | 58.24% | 61.25% | 63.98% | |
| Normal | 22.76% | 23.17% | 22.98% | 22.15% | |
| Low | 16.05% | 18.59% | 15.77% | 13.87% | |
| BMI | | | | | <0.001 |
| Normal | 28.73% | 24.68% | 29.74% | 31.63% | |
| Obese | 36.57% | 45.22% | 36.03% | 28.75% | |
| Overweight | 33.08% | 28.71% | 32.48% | 37.88% | |
| Underweight | 1.63% | 1.39% | 1.74% | 1.74% | |
| WBC (1000 cells/μl) | 7.27 (2.43) | 7.73 (2.74) | 7.21 (2.20) | 6.88 (2.26) | <0.001 |
| Neu (1000 cells/μl) | 4.30 (1.70) | 4.58 (1.79) | 4.25 (1.57) | 4.08 (1.69) | <0.001 |
| PLT (1000 cells/μl) | 243.57 (61.78) | 252.19 (64.54) | 246.68 (61.57) | 232.19 (57.43) | <0.001 |
| Lym (1000 cells/μl) | 2.16 (1.35) | 2.30 (1.70) | 2.16 (1.12) | 2.01 (1.15) | <0.001 |
| SII (1000 cells/μl) | 530.03 (290.47) | 543.63 (291.31) | 535.11 (298.91) | 511.92 (280.24) | <0.001 |
| Creatinine (μmol/L) | 78.48 (37.30) | 74.72 (28.91) | 78.80 (46.49) | 81.77 (33.80) | <0.001 |
| ALT (U/L) | 25.65 (17.19) | 23.74 (17.47) | 25.20 (16.04) | 27.95 (17.76) | <0.001 |
| AST(U/L) | 26.03 (16.82) | 24.66 (20.29) | 25.33 (11.13) | 28.05 (17.61) | <0.001 |
| ALP (IU/L) | 66.45 (21.63) | 68.17 (21.94) | 66.44 (22.46) | 64.79 (20.33) | <0.001 |
| Na (mmol/L) | 139.23 (2.24) | 139.19 (2.24) | 139.24 (2.21) | 139.24 (2.27) | 0.694 |
| K (mmol/L) | 3.97 (0.32) | 3.97 (0.33) | 3.97 (0.33) | 3.98 (0.31) | 0.859 |
| Smoking status | | | | | 0.340 |
| Yes | 44.98% | 44.83% | 46.59% | 43.54% | |

*(Continued)*

**Table 1.** (Continued)

| Characteristics | Overall, N = 5831 | TB Tertiles | | | p-value[2] |
|---|---|---|---|---|---|
| | | Q1, N = 1982[1] | Q2, N = 1921[1] | Q3, N = 1928[1] | |
| No | 55.02% | 55.17% | 53.41% | 56.46% | |
| Drinking status | | | | | 0.002 |
| Yes | 77.71% | 74.26% | 78.36% | 80.40% | |
| No | 22.29% | 25.74% | 21.64% | 19.60% | |
| Hypertension | | | | | 0.066 |
| Yes | 31.53% | 32.33% | 33.08% | 29.23% | |
| No | 68.47% | 67.67% | 66.92% | 70.77% | |
| Diabetes | | | | | <0.001 |
| Yes | 9.11% | 11.49% | 9.22% | 6.72% | |
| No | 90.89% | 88.51% | 90.78% | 93.28% | |

BFRs, brominated flame retardants; TB, total bilirubin; PIR, poverty-to-income ratio; BMI, body mass index; WBC, white blood cell; Neu, neutrophil count; PLT, platelet count; Lym, lymphocyte count; SII, systemic immune-inflammation index; ALT, alanine aminotransferase; AST, aspartate aminotransferase; ALP, Alkaline Phosphatase; Na, blood sodium level; K, blood potassium level. Q1-Q3 were grouped into study participants based on the tertile range of TB level (Q1, the first tertile; Q2, the second tertile; Q3: the third tertile).

[1]Mean (SD); %

[2]Wilcoxon rank-sum test for complex survey samples; chi-squared test with Rao & Scott's second-order correction.

to tertiles of TB levels, PBB153, PBDE17, PBDE47, PBDE85, PBDE99, PBDE100, PBDE183, and PBDE209, were significantly correlated with TB ($p < 0.05$). Additionally, factors such as gender, race, PIR, education level, BMI, WBC, Plt, Neu, Lym, SII, creatinine, ALT, AST, ALP, drinking status, and diabetes were also significantly correlated with TB ($p < 0.05$).

## 3.2 Association between BFRs and TB levels

After the log transformation of the concentrations of individual BFRs, it was found that BFRs and TB data did not follow a normal distribution. Therefore, Spearman's rank correlation analysis was used to assess the relationship between individual BFRs and TB. As shown in Table 2, most BFRs except for PBDE66 were positively correlated with TB levels, with statistical significance ($p < 0.05$). Specifically, PBDE66 showed a negative correlation with TB levels ($\rho = -0.372$, $p < 0.001$). PBDE183 had the highest Spearman rank correlation coefficient of 0.292.

Weighted linear regression analysis was conducted to investigate the relationship between log-transformed individual BFRs and TB levels in Table 3. Three models were developed, in Model 1, most BFRs, except PBDE66 (β: 0.459, 95% CI: - 0.063, 0.980, $p = 0.084$), showed a statistically significant positive association with TB levels ($p < 0.05$). After adjusting for confounding factors, most BFRs continued to have a significant positive association with TB levels. However, PBDE153 (β: - 0.031, 95% CI: - 0.317, 0.255, $p = 0.829$) and PBDE66 (β: 0.285, 95% CI: - 0.208, 0.777, $p = 0.253$) did not show statistically significant associations.

In addition, based on Model 3, RCS was used to explore the nonlinear relationships between log-transformed individual BFRs and TB levels. As shown in Fig 2, PBDE153 exhibited a U-shaped significant association with TB levels ($p < 0.05$). PBDE17, PBDE99, PBDE154, and PBDE209 demonstrated an inverse J-shaped relationship with TB levels ($p < 0.05$). PBB153, PBDE66, PBDE85, and PBDE183 showed a similar S-shaped significant nonlinear association with TB levels ($p < 0.05$). In contrast, the nonlinear associations between PBDE28, PBDE47, and PBDE100 with TB levels were not significant ($p = 0.776$, $p = 0.085$, $p = 0.375$).

In Tables 4 and 5, stratification by age revealed an interaction between PBDE153 and TB levels (p for interaction = 0.012). Stratification by sex showed interactions between PBDE17 (p for interaction = 0.017) and PBDE183 (p for interaction = 0.027) with TB levels. No

**Table2. The associations between log-transformed serum BFRs and TB level by Spearman rank correlation analyses.**

| Log BFRs | ρ | p-value |
|---|---|---|
| PBB153 | 0.113 | <0.001 |
| PBDE17 | 0.260 | <0.001 |
| PBDE28 | 0.234 | <0.001 |
| PBDE47 | 0.264 | <0.001 |
| PBDE66 | -0.372 | <0.001 |
| PBDE85 | 0.215 | <0.001 |
| PBDE99 | 0.254 | <0.001 |
| PBDE100 | 0.238 | <0.001 |
| PBDE153 | 0.037 | 0.005 |
| PBDE154 | 0.252 | <0.001 |
| PBDE183 | 0.292 | <0.001 |
| PBDE209 | 0.198 | <0.001 |

BFRs, brominated flame retardants; TB, total bilirubin; $p < 0.05$ was considered statistically significant.

**Table 3. The associations between log-transformed serum BFRs and TB level by linear regression analyses.**

| | Model 1 | | Model 2 | | Model 3 | |
|---|---|---|---|---|---|---|
| Log BFRs | β (95%CI) | p-value | β (95%CI) | p-value | β (95%CI) | p-value |
| PBB153 | 0.560(0.399, 0.721) | <0.001 | 0.429(0.259, 0.598) | <0.001 | 0.460(0.288, 0.631) | <0.001 |
| PBDE17 | 2.170(1.699, 2.641) | <0.001 | 2.087(1.672, 2.503) | <0.001 | 2.083(1.673, 2.493) | <0.001 |
| PBDE28 | 1.276(0.905, 1.647) | <0.001 | 1.411(1.039, 1.783) | <0.001 | 1.406(1.037, 1.776) | <0.001 |
| PBDE47 | 1.135(0.802, 1.469) | <0.001 | 1.098(0.779, 1.416) | <0.001 | 1.090(0.775, 1.405) | <0.001 |
| PBDE66 | 0.459(-0.063, 0.980) | 0.084 | 0.301(-0.196, 0.797) | 0.232 | 0.285(-0.208, 0.777) | 0.253 |
| PBDE85 | 0.721(0.413, 1.029) | <0.001 | 0.687(0.396, 0.978) | <0.001 | 0.674(0.387, 0.961) | <0.001 |
| PBDE99 | 0.886(0.596, 1.175) | <0.001 | 0.838(0.562, 1.115) | <0.001 | 0.831(0.558, 1.104) | <0.001 |
| PBDE100 | 0.954(0.651, 1.258) | <0.001 | 0.894(0.619, 1.170) | <0.001 | 0.886(0.614, 1.157) | <0.001 |
| PBDE153 | 0.427(0.124, 0.731) | 0.006 | -0.033(-0.327, 0.261) | 0.822 | -0.031(-0.317, 0.255) | 0.829 |
| PBDE154 | 0.973(0.692, 1.254) | <0.001 | 0.938(0.692, 1.85) | <0.001 | 0.927(0.682, 1.172) | <0.001 |
| PBDE183 | 1.144(0.801, 1.487) | <0.001 | 0.787(0.467, 1.107) | <0.001 | 0.774(0.455, 1.109) | <0.001 |
| PBDE209 | 1.415(1.055, 1.878) | <0.001 | 1.020(0.695, 1.344) | <0.001 | 1.021(0.697, 1.346) | <0.001 |

BFRs, brominated flame retardants; TB, total bilirubin; PIR, poverty-to-income ratio; BMI, body mass index; 95% CI, 95% confidence interval; $p < 0.05$ was considered statistically significant.

M1: Unadjusted;

M2: Adjust for age, gender, race, PIR, education, and BMI;

M3: Adjust for age, gender, race, PIR, education, BMI, smoking status, drinking status, hypertension and diabetes.

significant interactions were observed for other BFRs ($p > 0.05$). Additionally, PBDE66 and PBDE153 did not show statistically significant associations with TB ($p > 0.05$), whereas other BFRs exhibited significant differences ($p < 0.05$). Thus, a significant positive correlation was found between most BFRs and TB levels.

### 3.3 Association between mixed BFRs exposure and TB levels

After adjusting for confounding factors (Model 3), the relationship between mixed BFRs exposure and TB levels was analyzed using WQS regression and QGC analysis. As shown in

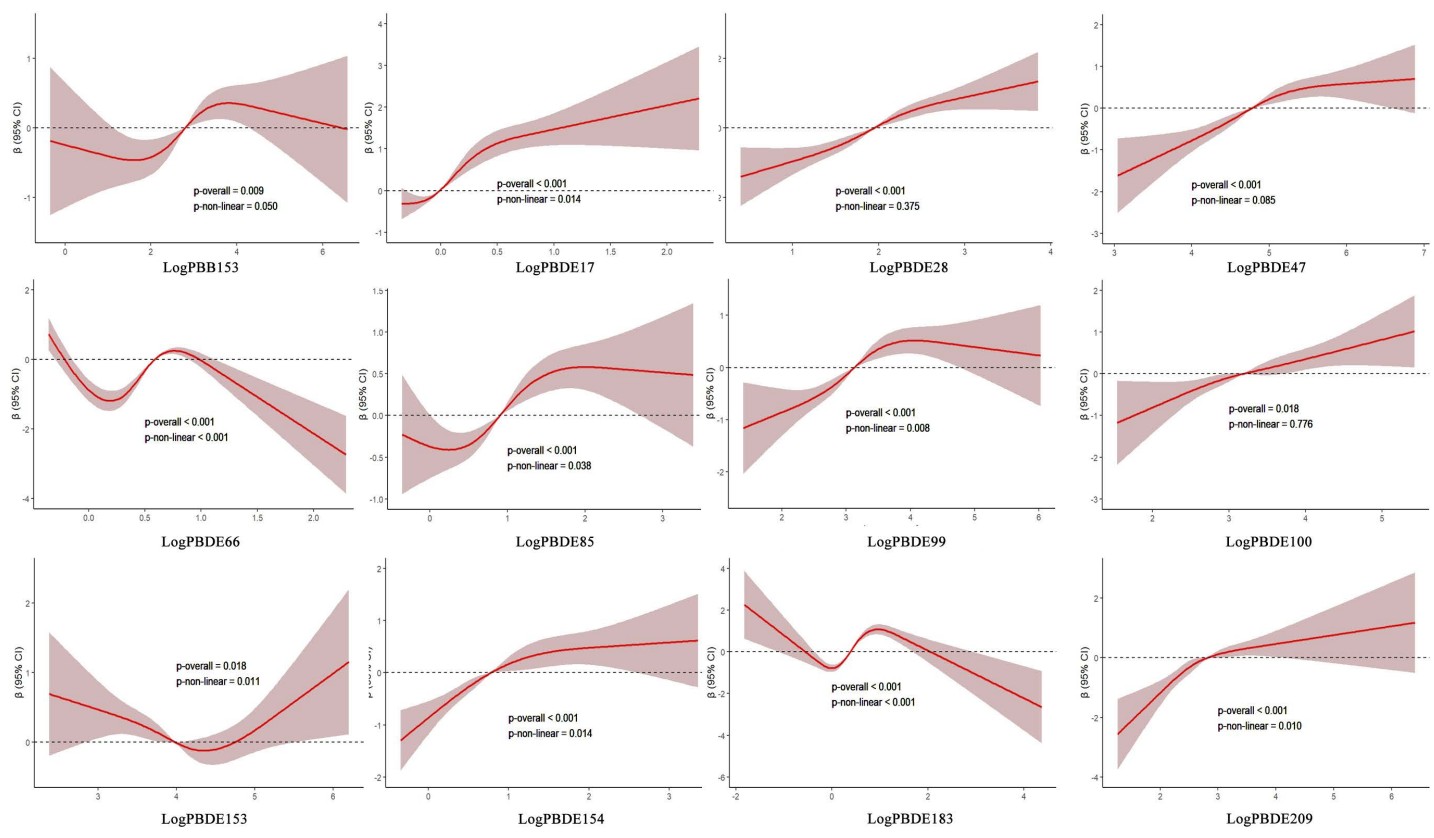

**Fig 2. Nonlinear associations of log BFRs and TB.** (BFRs, brominated flame retardants; TB, total bilirubin).

Fig 3, the WQS coefficient for mixed BFRs exposure was positively associated with TB levels (β: 0.553, 95% CI: 0.384, 0.722, $p < 0.001$), with PBDE183 contributing the most at 54.8% (Fig 3A). The QGC analysis corroborated the WQS regression results, indicating that exposure to mixed BFRs was positively associated with TB levels (β: 1.060, 95% CI: 0.587, 1.532, $p < 0.001$), with PBDE183 remaining the most significant contributor, accounting for 31.2%.

## 4. Discussion

This study demonstrates a positive correlation between BFRs and TB levels. Our findings indicate that most BFRs analyzed were significantly positively correlated with TB levels ($p < 0.05$), with PBDE183 having the highest Spearman rank correlation coefficient of 0.292. After adjusting for confounding factors, weighted linear regression analysis revealed that, except for PBDE66 and PBDE153, which did not show statistically significant associations with TB ($p > 0.05$), other BFRs exhibited a significant positive correlation with TB ($p < 0.05$). Additionally, nonlinear correlation analysis revealed that PBDE28, PBDE47, and PBDE100 did not show significant nonlinear associations with TB levels, while other BFRs demonstrated significant nonlinear relationships. WQS regression and QGC analysis confirmed a positive association between exposure to mixed BFRs and TB levels (β: 1.060, 95% CI: 0.587, 1.532, $p < 0.001$), with PBDE183 contributing the most.

After human exposure to BFRs, these substances are distributed in various tissues such as serum, placenta, breast milk, liver, and fat. This distribution can further impact the endocrine system, nervous system, reproductive system, and liver metabolism, among other physiological

**Table 4. Association between log-transformed serum BFRs and TB stratified by age.**

| Log BFRs | Subgroup | β (95%CI) | p-value | p for interaction |
|---|---|---|---|---|
| PBB153 | <65 | 0.370(0.189, 0.552) | <0.001 | 0.924 |
|  | ≥65 | 0.427(0.075, 0.779) | 0.018 |  |
| PBDE17 | <65 | 1.949(1.555, 2.344) | <0.001 | 0.330 |
|  | ≥65 | 2.115(1.564, 2.667) | <0.001 |  |
| PBDE28 | <65 | 1.504(1.092, 1.917) | <0.001 | 0.306 |
|  | ≥65 | 0.681(0.174, 1.189) | 0.009 |  |
| PBDE47 | <65 | 1.245(0.866, 1.624) | <0.001 | 0.260 |
|  | ≥65 | 0.424(-0.047, 0.895) | 0.077 |  |
| PBDE66 | <65 | 0.443(-0.162, 1.049) | 0.149 | 0.375 |
|  | ≥65 | -0.314(-1.083, 0.455) | 0.418 |  |
| PBDE85 | <65 | 0.769(0.413, 1.126) | <0.001 | 0.487 |
|  | ≥65 | 0.337(-0.034, 0.707) | 0.074 |  |
| PBDE99 | <65 | 0.970(0.637, 1.302) | <0.001 | 0.149 |
|  | ≥65 | 0.286(-0.088, 0.660) | 0.131 |  |
| PBDE100 | <65 | 1.028(0.690, 1.365) | <0.001 | 0.421 |
|  | ≥65 | 0.299(-0.151, 0.750) | 0.189 |  |
| PBDE153 | <65 | 0.079(-0.232, 0.390) | 0.613 | 0.012 |
|  | ≥65 | -0.414(-0.931, 0.103) | 0.114 |  |
| PBDE154 | <65 | 1.034(0.720, 1.347) | <0.001 | 0.284 |
|  | ≥65 | 0.538(0.116, 0.960) | 0.013 |  |
| PBDE183 | <65 | 0.678(0.323, 1.033) | <0.001 | 0.073 |
|  | ≥65 | 1.400(0.964, 1.836) | <0.001 |  |
| PBDE209 | <65 | 1.008(0.614, 1.403) | <0.001 | 0.932 |
|  | ≥65 | 1.174(0.406, 1.942) | 0.003 |  |

BFRs, brominated flame retardants; TB, total bilirubin; 95% CI, 95% confidence interval; $p < 0.05$ was considered statistically significant.

processes, ultimately exerting negative effects on human health [25–32]. Studies have shown that higher levels of PBDEs are associated with longer time to conception in women and can impact newborns' weight, head circumference, and length [30,33]. Additionally, exposure to PBDEs can affect preschool children's attention, fine motor coordination, cognitive abilities, intelligence, physical development, and so on [30,34]. Additionally, thyroid hormone levels can impact female reproductive health, and studies have shown that exposure to BFRs is associated with thyroid hormone levels [27,35,36]. Harley et al. have suggested that BFRs such as "PBDE 28, 47, 100, 153, 190" may affect neurological development and reproductive systems by disrupting thyroid hormone homeostasis [13]. Exposure to flame retardants has also been associated with respiratory diseases such as chronic obstructive pulmonary disease (COPD) and pulmonary ventilatory function, with the largest contributors being PBB153 and PBDE47, respectively, for both individual and mixtures of BFRs [37,38]. And exposure to a mixture of BFRs was also positively associated with adult metabolic syndrome (MetS) and its components, with PBB153, PBDE28, and PBDE209 being important chemicals [39]. However, there is still insufficient epidemiological evidence to establish definitive causal relationships between BFRs exposure and these health outcomes, highlighting the need for more research to elucidate these connections.

Bilirubin has various functions that influence biological activities, ranging from cell signaling and metabolic regulation to immune modulation, and it exhibits notable clinical and even therapeutic effects [40]. However, elevated bilirubin levels can lead to neurotoxicity, and damage to organs such as the heart, liver, and kidneys, which is harmful to human health

**Table 5. Association between log-transformed serum BFRs and TB stratified by gender.**

| Log BFRs | Subgroup | β (95%CI) | *p*-value | p for interaction |
|---|---|---|---|---|
| **PBB153** | male | 0.619(0.367, 0.871) | <0.001 | 0.993 |
| | female | 0.280(0.056, 0.505) | 0.015 | |
| **PBDE17** | male | 2.786(2.012, 3.560) | <0.001 | 0.017 |
| | female | 1.488(1.079, 1.897) | <0.001 | |
| **PBDE28** | male | 1.672(1.115, 2.229) | <0.001 | 0.367 |
| | female | 1.148(0.688, 1.608) | <0.001 | |
| **PBDE47** | male | 1.315(0.848, 1.782) | <0.001 | 0.257 |
| | female | 0.856(0.463, 1.249) | <0.001 | |
| **PBDE66** | male | 0.472(-0.138, 1.082) | 0.127 | 0.516 |
| | female | 0.063(-0.696, 0.823) | 0.868 | |
| **PBDE85** | male | 0.882(0.448, 1.315) | <0.001 | 0.228 |
| | female | 0.465(0.119, 0.810) | 0.009 | |
| **PBDE99** | male | 1.050(0.651, 1.449) | <0.001 | 0.894 |
| | female | 0.607(0.266, 0.947) | <0.001 | |
| **PBDE100** | male | 1.041(0.618, 1.465) | <0.001 | 0.408 |
| | female | 0.713(0.372, 1.055) | <0.001 | |
| **PBDE153** | male | 0.030(-0.421, 0.481) | 0.896 | 0.775 |
| | female | -0.139(-0.429, 0.150) | 0.339 | |
| **PBDE154** | male | 1.139(0.742, 1.537) | <0.001 | 0.152 |
| | female | 0.683(0.411, 0.955) | <0.001 | |
| **PBDE183** | male | 1.038(0.583, 1.493) | <0.001 | 0.027 |
| | female | 0.367(0.026, 0.708) | 0.035 | |
| **PBDE209** | male | 1.097(0.629, 1.565) | <0.001 | 0.619 |
| | female | 0.874(0.405, 1.343) | <0.001 | |

BFRs, brominated flame retardants; TB, total bilirubin; 95% CI, 95% confidence interval; *p* < 0.05 was considered statistically significant.

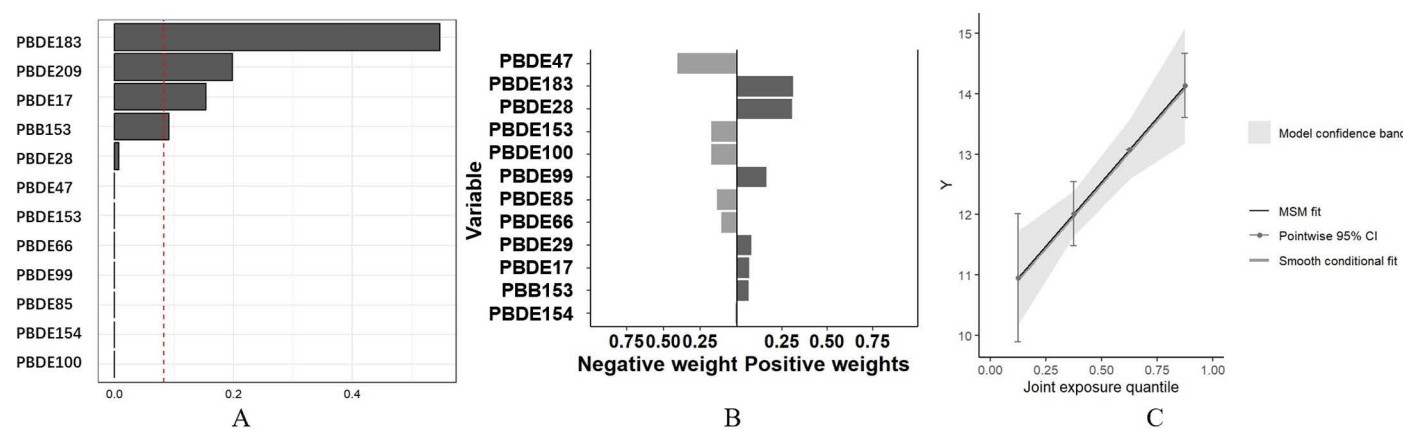

**Fig 3. Associations of mixture BFRs exposure with TB level in the study participants.** (A) WQS regression index weights for TB level. (B) QGC regression index weights for TB level. (C) Joint effect (95% CI) for TB level in QGC analysis. (BFRs, brominated flame retardants; TB, total bilirubin; 95% CI, 95% confidence interval).

[21,41–43]. Several studies have shown that elevated TB levels are associated with the severity of coronary artery disease in non-ST-segment elevation myocardial infarction patients [44], an increased risk of adverse cardiovascular events [45], prolonged QT interval [46], and a higher risk of cardiovascular disease (CVD) and overall mortality [47]. A study including 1,732 healthy males found that indirect bilirubin levels are an independent risk factor for elevated cholesterol [48]. Another retrospective study indicated a positive correlation between TB levels and acute exacerbations of COPD [49]. These results suggest that elevated bilirubin levels could present potential health risks. However, Spearman rank correlation analyses suggested that the correlations between individual BFRs and TB were all weak, which may be related to the sample size of the study, thus necessitating a large sample size. In addition, after adjusting for confounders, the association between PBDE153 and TB was no longer statistically significant, the reason for which needs to be further explored.

Previous studies have shown that bilirubin levels increase with age, and that men have higher bilirubin levels than women [50,51]. And environmental factors also contribute to changes in specific processes of aging [52]. Therefore, the present study was stratified for age and gender after adjusting for confounding factors. The results still suggested that most of the BFRs were positively associated with TB levels and the association of PBDE66 and PBDE153 with TB was not statistically significant. This suggests that the results of this study are robust and that exposure to BFRs may lead to elevated levels of TB, which may have systemic effects that threaten human health.

However, the study has several limitations. First, the various pathways of BFRs exposure could not be determined to ascertain which pathway has the most significant impact on TB levels. Second, the study focused only on the association between TB levels and exposure to BFRs, highlighting the need for further research into direct and indirect bilirubin. Additionally, the study utilized data from a single measurement; using mean values from multiple measurements might yield more accurate results. Lastly, despite including numerous covariates, there may still be other confounding factors influencing the results. Therefore, future research with larger sample sizes is needed to more comprehensively elucidate the relationship between BFRs exposure and TB levels.

## 5. Conclusion

This study confirms a significant positive correlation between BFRs and TB levels, further suggesting potential adverse effects of BFRs exposure on various human systems. The mechanisms underlying these effects require more extensive and detailed research.

## Acknowledgements

The authors express their gratitude to all participants of the NHANES.

## Author contributions

**Conceptualization:** Tong Lin, Jialu Chen, Fen Zhou, Junjie Yang, Haiyan Mao, Zhouxin Yang.

**Data curation:** Shanshan Huang, Tong Lin, Zhouxin Yang.

**Formal analysis:** Jialu Chen, Zhouxin Yang.

**Funding acquisition:** Fen Zhou, Zhouxin Yang.

**Investigation:** Shanshan Huang, Tong Lin, Haiyan Mao, Zhouxin Yang.

**Methodology:** Shanshan Huang, Jialu Chen, Fen Zhou, Zhouxin Yang.

**Project administration:** Fen Zhou.

**Resources:** Junjie Yang, Haiyan Mao.

**Software:** Shanshan Huang, Jialu Chen, Zhouxin Yang.

**Supervision:** Tong Lin, Fen Zhou, Junjie Yang, Zhouxin Yang.

**Validation:** Junjie Yang, Haiyan Mao.

**Visualization:** Junjie Yang.

**Writing – original draft:** Shanshan Huang, Haiyan Mao.

**Writing – review & editing:** Haiyan Mao, Zhouxin Yang.

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
