## [Decision Letter · Decision Letter 0]

16 Jan 2025

PONE-D-24-39826The Impact of Brominated Flame Retardants Exposure on Serum Total Bilirubin: A Cross-Sectional AnalysisPLOS ONE

Dear Dr. Mao,

Thank you for submitting your manuscript to PLOS ONE. After careful consideration, we feel that it has merit but does not fully meet PLOS ONE’s publication criteria as it currently stands. Therefore, we invite you to submit a revised version of the manuscript that addresses the points raised during the review process.

We look forward to receiving your revised manuscript.

Kind regards,

Satish Rojekar, Ph.D.

Academic Editor

PLOS ONE

**Journal Requirements:**

Zhejiang Provincial Natural Science Foundation of China (LY21H150002) and Scientific Research Fund of Zhejiang Province Education Department (Y202043331).

Reviewers' comments:

Reviewer's Responses to Questions

**Comments to the Author**

1. Is the manuscript technically sound, and do the data support the conclusions?

Reviewer #1: Yes

Reviewer #2: Partly

Reviewer #3: Yes

2. Has the statistical analysis been performed appropriately and rigorously? 

Reviewer #1: Yes

Reviewer #2: Yes

Reviewer #3: Yes

3. Have the authors made all data underlying the findings in their manuscript fully available?

Reviewer #1: Yes

Reviewer #2: Yes

Reviewer #3: Yes

4. Is the manuscript presented in an intelligible fashion and written in standard English?

Reviewer #1: Yes

Reviewer #2: Yes

Reviewer #3: Yes

5. Review Comments to the Author

**Reviewer #1: ** While the paper presents valuable insights, I recommend some revisions to enhance its clarity and impact:

1. “According to the World Health Organization, in 2016, ecological pollution was responsible for an estimated 4.2 million deaths globally”. How does this related to statistics for deaths globally due to BFRs? State accurate statistics to help understand the importance of study.

2. Explain how humans are exposed to BFRs in a day-to-day life to help understand the toxic effect of BFRs.

3. The initial dataset had 50588 participants, however after filtering only 5831 participants were left which is approximately 11% of the total participants. Does this mean that association between BFRs and total bilirubin is not an important parameter measured while recording data in NHANES?

**Reviewer #2:**  Comments for the Research paper entitled ‘The Impact of Brominated Flame Retardants Exposure on Serum Total Bilirubin: A Cross-Sectional Analysis’

1. What is the significance of this study in the context of comprehending the relationship between BFRs and total bilirubin (TB) levels?

2. Provide additional information regarding the brominated flame retardants (BFRs) and the reasons for their classification as hazardous environmental pollutants in

the introduction section.

3. In this investigation, how were total bilirubin (TB) levels classified?

4. Check the discrepancy between the years of NHANES data mentioned in the

abstract (2007–2016) and the methodology (2005–2016).

5. The methodology for measuring BFRs and TB levels is unclear. Provide/cite the relevant references or links to NHANES protocols to enhance clarity.

6. How did adjusting for confounding factors affect the results?

7. What types of correlations (e.g., U-shaped, inverted J shaped) were observed between specific BFRs and TB levels and what is the significance of these findings?

8. What was the impact of the stratified analysis by gender and age on the results?

9. In what manner were combined exposures to BFRs assessed in this study?

10. What is the significance of stratified analysis by age and gender in understanding the correlation between BFRs and TB levels?

11. In the discussion section, provide additional information regarding the significant PBDEs observed in conditions such as metabolic syndrome, COPD and neurological disorders.

**Reviewer #3: ** The authors have performed the study in a well organized manner and following are my comments for them -

1) please check for the typos and grammatical errors.

2) In table 3, for linear regression analysis, please correct the spelling for 'model 'in the header

3) In the findings for the study, the highest Spearman ranking was 0.292 with p-value <0.05. Usually, the correlation value <0.4 is considered as weak correlation. I would ask the authors to expand on the findings with weak correlation and what further investigation can be done to improve it.

6. PLOS authors have the option to publish the peer review history of their article (what does this mean? ). If published, this will include your full peer review and any attached files.

**Do you want your identity to be public for this peer review?** For information about this choice, including consent withdrawal, please see our Privacy Policy .

Reviewer #1: No

Reviewer #2: No

Reviewer #3: No

---

## [Author Response · Author response to Decision Letter 1]

13 Feb 2025

Dear Editors,

Thank you for your comments concerning our manuscript entitled “ The impact of brominated flame retardants exposure on serum total bilirubin: A cross-sectional analysis ”. Your comments are all precious and helpful in the revision and improvements of our paper and they also have crucial guiding significance to our research, as well. We have studied your comments carefully and have made corrections that we hope can meet expectations. Revised portions are marked in red on the paper. The main corrections in the paper and the responses to your comments are as follows:

Reviewers’ reports:

Reviewer #1: While the paper presents valuable insights, I recommend some revisions to enhance its clarity and impact:

Question 1: “According to the World Health Organization, in 2016, ecological pollution was responsible for an estimated 4.2 million deaths globally”. How does this related to statistics for deaths globally due to BFRs? State accurate statistics to help understand the importance of study.

Response: Thank you for your valuable comment. According to your comments, we have added “Although there is currently no statistical data on deaths caused by environmental pollution from brominated flame retardants (BFRs), evidence suggests that exposure to BFRs is closely associated with adverse health outcomes and may increase the risk of tumor-related mortality [2].” to the fourth line of the first paragraph of the new manuscript for better understanding.

Question 2: Explain how humans are exposed to BFRs in a day-to-day life to help understand the toxic effect of BFRs.

Response: We sincerely appreciate the valuable comments. Based on your comments, we have added a relevant section to the second paragraph of the new manuscript on how humans are exposed to brominated flame retardants (BFRs) in their daily lives, as follows “They are widely used in commercial and industrial applications, including furniture, textiles, plastics, electronic devices, and building materials, to reduce flammability and decrease the risk of fires, meeting fire safety standards. Humans are continuously exposed to these flame retardants through various routes, including diet, breastfeeding, drinking water, and using products containing these substances [4–6]. Despite the banning of several flame retardants, these substances persist in the environment due to their slow degradation and tendency to bioaccumulate [7–9]. It has been reported that although the United States ceased the production of Polychlorinated Biphenyls (PCBs) in 1979, these substances remain persistent in the environment and can accumulate through breastfeeding and the food chain [10–12].”.

Question 3: The initial dataset had 50588 participants, however after filtering only 5831 participants were left which is approximately 11% of the total participants. Does this mean that association between BFRs and total bilirubin is not an important parameter measured while recording data in NHANES?

Response: We think this is an excellent suggestion. We fully agree with your considerations regarding the final inclusion of 11% of the total investigator population. However, because the NHANES database records of BFRs concentrations use mixed samples, the samples are pooled before analytical measurements are made. While this expands the sample size and lowers the limit of detection, the reduction in the number of measurements and cost allows for larger sample volumes. However, documentation of BFR concentrations is still missing. Thus, although the total study population sample size was 50,588, the number of participants was 8,836 after removing the missing data on BFRs, while the number of missing total bilirubin was 20, and 5,831 covariates were excluded from the study, so the difference in missing data may not be significant, but it is a reminder to be aware of this issue in future studies.。

Reviewer #2: Comments for the Research paper entitled ‘The Impact of Brominated Flame Retardants Exposure on Serum Total Bilirubin: A Cross-Sectional Analysis’

Question 1: What is the significance of this study in the context of comprehending the relationship between BFRs and total bilirubin (TB) levels?

Response: Thank you for your valuable comment. In the context of comprehending the relationship between BFRs and total bilirubin (TB) level, the significance of this study is that the exposure to BFRs was found to be positively correlated with total bilirubin concentration, suggesting that BFRs may affect the body's immune system, digestive system, and inflammatory response system, among others, by elevating bilirubin. This suggests that exposure to BFRs may lead to elevated bilirubin levels, which in turn increases the risk of organ damage. The potential threat of BFRs as environmental pollutants to public health was also emphasized.

Question 2: Provide additional information regarding the brominated flame retardants (BFRs) and the reasons for their classification as hazardous environmental pollutants in the introduction section.

Response: We think this is an excellent suggestion. Based on your comments, we have added more information on brominated flame retardants in the second paragraph of the new manuscript. Specifically, “They are widely used in commercial and industrial applications, including furniture, textiles, plastics, electronic devices, and building materials, to reduce flammability and decrease the risk of fires, meeting fire safety standards. Humans are continuously exposed to these flame retardants through various routes, including diet, breastfeeding, drinking water, and using products containing these substances [4–6]. Despite the banning of several flame retardants, these substances persist in the environment due to their slow degradation and tendency to bioaccumulate [7–9]. It has been reported that although the United States ceased the production of Polychlorinated Biphenyls (PCBs) in 1979, these substances remain persistent in the environment and can accumulate through breastfeeding and the food chain [10–12].”.

Question 3: In this investigation, how were total bilirubin (TB) levels classified?

Response: Thank you for your valuable comment. We apologize for not clarifying the classification of total bilirubin. Total bilirubin is categorized by tertiles. In order to clarify the classification of total bilirubin, we have added a note on Q1-Q3 in the notes to Table 1, i.e., “Q1-Q3 were grouped into study participants based on the tertile range of TB level (Q1, the first tertile; Q2, the second tertile; Q3: the third tertile).”.

Question 4: Check the discrepancy between the years of NHANES data mentioned in the

abstract (2007–2016) and the methodology (2005–2016).

Response: We sincerely appreciate the valuable comments. We apologize for the inconsistency in the years in the abstract and methods sections. We have carefully checked the data of this study and determined that the years of this study are 2007-2016 and have revised the methods section of the new manuscript to read 2007-2016.

Question 5: The methodology for measuring BFRs and TB levels is unclear. Provide/cite the relevant references or links to NHANES protocols to enhance clarity.

Response: We think this is an excellent suggestion. We apologize for not clarifying the measurement methods for brominated flame retardants and total bilirubin. In the new Methods section of the manuscript, our measurement methods for brominated flame retardants and total bilirubin have been rewritten to read “BFRs were analyzed using automated liquid-liquid extraction and purification, followed by target analyte analysis through isotope dilution gas chromatography with high-resolution mass spectrometry GC/IDHRMS [23]. TB was measured and quality-controlled using the timed-endpoint diazo method (Jendrassik-Grof) at the NHANES mobile examination centers [24].”.

Question 6: How did adjusting for confounding factors affect the results?

Response: Thank you for your valuable comment. After adjusting for confounding factors, the association between PBDE153 and total bilirubin was no longer statistically significant, while the other BFRs remained positively associated with total bilirubin. In the third paragraph of the Discussion section of the new manuscript, we added “In addition, after adjusting for confounders, the association between PBDE153 and TB was no longer statistically significant, the reason for which needs to be further explored.”.

Question 7: What types of correlations (e.g., U-shaped, inverted J shaped) were observed between specific BFRs and TB levels and what is the significance of these findings?

Response: Thank you for your valuable comment. In Figure 2, PBDE153 exhibited a U-shaped significant association with TB levels (p < 0.05). PBDE17, PBDE99, PBDE154, and PBDE209 demonstrated an inverse J-shaped relationship with TB levels (p < 0.05). PBB153, PBDE66, PBDE85, and PBDE183 showed a similar S-shaped significant nonlinear association with TB levels (p < 0.05). The U-shape shows that exposure levels at low and high concentrations have a more significant effect on the results, while intermediate values have a lesser effect, and is similar in shape to a “U”. The S-shape shows a low effect at low concentrations, a gradual increase at intermediate concentrations, and a higher effect at high concentrations, with a greater emphasis on the significant effect of intermediate values, and is close in shape to the S. The inverted J-shape suggests that exposure levels have the most significant effect on results within a certain concentration range, beyond which exposure levels have a lesser effect on the results. The inverted J-shape suggests that exposure levels have the most significant effect on the results within a certain concentration range, beyond which exposure levels have less effect on the results.

Question 8: What was the impact of the stratified analysis by gender and age on the results?

Response: We sincerely appreciate the valuable comments. Stratified analyses for gender and age still suggested that most of the BFRs were positively associated with total bilirubin levels, but the association of PBDE66 and PBDE153 with bilirubin was not statistically significant. Suggesting that the results of this study are robust.

Question 9: In what manner were combined exposures to BFRs assessed in this study?

Response: Thank you for your valuable comment. In this study, we used WQS regression and QGC analysis to examine the overall effect of BRFs exposure on total bilirubin. WQS regression models are commonly used to explore the directional, linear, and cumulative effects of mixed chemical exposures on human health outcomes [1]. And the training set (30%) and validation set (70%) were randomly assigned in the model. The overall effect of BFR was investigated by calculating regression coefficients using 1,000 bootstrap samples from the training dataset. A QGC analysis was also performed, combining the inference simplicity of WQS regression with the flexibility of g calculations to assess the cumulative effect of BRF on total bilirubin [2].

[1] Carrico C, Gennings C, Wheeler DC, Factor-Litvak P. Characterization of weighted quantile sum regression for highly correlated data in a risk analysis setting. J Agric Biol Environ Stat. (2015) 20:100–20. 10.1007/s13253-014-0180-3.

[2] Keil AP, Buckley JP, O'Brien KM, Ferguson KK, Zhao S, White AJ. A quantile-based g-computation approach to addressing the effects of exposure mixtures. Environ Health Perspect. (2020) 128:047004. 10.1289/EHP5838.

Question 10: What is the significance of stratified analysis by age and gender in understanding the correlation between BFRs and TB levels?

Response: We sincerely appreciate the valuable comments. Previous studies have shown that bilirubin levels increase with age, and that men have higher bilirubin levels than women [1,2]. And environmental factors also contribute to changes in specific processes of aging [3]. Therefore, in order to explore whether gender and age affect the positive correlation between brominated flame retardants and total bilirubin, we performed stratified analyses to better illustrate the association between brominated flame retardants exposure and total bilirubin. And we have added this section to paragraph 4 of the Discussion section of the new manuscript, as follows “Previous studies have shown that bilirubin levels increase with age, and that men have higher bilirubin levels than women [50,51]. And environmental factors also contribute to changes in specific processes of aging [52]. Therefore, the present study was stratified for age and gender after adjusting for confounding factors. The results still suggested that most of the BFRs were positively associated with TB levels and the association of PBDE66 and PBDE153 with TB was not statistically significant. This suggests that the results of this study are robust and that exposure to BFRs may lead to elevated levels of TB, which may have systemic effects that threaten human health.”.

[1] Boland BS, Dong MH, Bettencourt R, Barrett-Connor E, Loomba R. Association of serum bilirubin with aging and mortality. J Clin Exp Hepatol. 2014 Mar;4(1):1-7. doi: 10.1016/j.jceh.2014.01.003.

[2] Lan Y, Liu H, Liu J, Zhao H, Wang H. The Relationship Between Serum Bilirubin Levels and Peripheral Arterial Disease and Gender Difference in Patients With Hypertension: BEST Study. Angiology. 2020 Apr;71(4):340-348. doi: 10.1177/0003319719900734.

[3] López-Otín C, Blasco MA, Partridge L, Serrano M, Kroemer G. The hallmarks of aging. Cell. 2013 Jun 6;153(6):1194-217. doi: 10.1016/j.cell.2013.05.039.

Question 11: In the discussion section, provide additional information regarding the significant PBDEs observed in conditions such as metabolic syndrome, COPD and neurological disorders.

Response: We think this is an excellent suggestion. Based on your comments, we have revised the Discussion section of the new manuscript to read “Harley et al. have suggested that BFRs such as “PBDE 28, 47, 100, 153, 190” may affect neurological development and reproductive systems by disrupting thyroid hormone homeostasis [13]. Exposure to flame retardants has also been associated with respiratory diseases such as chronic obstructive pulmonary disease (COPD) and pulmonary ventilatory function, with the largest contributors being PBB153 and PBDE47, respectively, for both individual and mixtures of BFRs [37,38]. And exposure to a mixture of BFRs was also positively associated with adult metabolic syndrome (MetS) and its components, with PBB153, PBDE28, and PBDE209 being important chemicals [39].”.

Reviewer #3: The authors have performed the study in a well organized manner and following are my comments for them -

Question 1: please check for the typos and grammatical errors.

Response: We sincerely appreciate the valuable comments. Based on your comments, we have carefully revised the grammar and misspelled words throughout the text.

Question 2: In table 3, for linear regression analysis, please correct the spelling for 'model 'in the header

Response: Thank you for your valuable comment. We apologize for the misspelling of the word model. In Table 3 of the new manuscript, we have changed “modle” to “model”.

Question 3: In the findings for the study, the highest Spearman ranking was 0.292 with p-value <0.05. Usually, the correlation value <0.4 is considered as weak correlation. I would ask the authors to expand on the findings with weak correlation and what further investigation can be done to improve it.

Response: We sincerely appreciate the valuable comments. We fully agree with your comments, so in the third paragraph of the Discussion section of the new manuscript, we have added “However, Spearman rank correlation analyses suggested that the correlations between individual BFRs and TB were all weak, which may be related to the sample size of the study, thus necessitating a large sample size. In addition, after adjusting for confounders, the association between PBDE153 and TB was no longer statistically significant, the reason for which needs to be further explored.”

---

## [Editor Report · Decision Letter 1]

20 Feb 2025

The impact of brominated flame retardants exposure on serum total bilirubin: A cross-sectional analysis

PONE-D-24-39826R1

Dear Dr. Mao,

We’re pleased to inform you that your manuscript has been judged scientifically suitable for publication and will be formally accepted for publication once it meets all outstanding technical requirements.

Kind regards,

Satish Rojekar, Ph.D.

Academic Editor

PLOS ONE

---

## [Editor Report · Acceptance letter]

PONE-D-24-39826R1

PLOS ONE

Dear Dr. Mao,

I'm pleased to inform you that your manuscript has been deemed suitable for publication in PLOS ONE. Congratulations! Your manuscript is now being handed over to our production team.

Kind regards,

on behalf of

Dr. Satish Rojekar

Academic Editor

PLOS ONE